# Ultra-High-Frequency Ultrasound as an Innovative Imaging Evaluation of Hyaluronic Acid Filler in Nasolabial Folds

**DOI:** 10.3390/diagnostics13172761

**Published:** 2023-08-25

**Authors:** Giorgia Salvia, Nicola Zerbinati, Flavia Manzo Margiotta, Alessandra Michelucci, Giammarco Granieri, Cristian Fidanzi, Riccardo Morganti, Marco Romanelli, Valentina Dini

**Affiliations:** 1Department of Dermatology, University of Pisa, 56126 Pisa, Italy; manzomargiottaflavia@gmail.com (F.M.M.); alessandra.michelucci@gmail.com (A.M.); giammarcogranieri@gmail.com (G.G.); cri.fidanzi@outlook.it (C.F.); romanellimarco60@gmail.com (M.R.); valentinadini74@gmail.com (V.D.); 2Dermatologic Unit, Department of Medicine and Surgery, University of Insubria, 21100 Varese, Italy; nicola.zerbinati@uninsubria.it; 3Section of Statistics, Department of Clinical and Experimental Medicine, University of Pisa, 56126 Pisa, Italy; r.morganti@ao-pisa.toscana.it

**Keywords:** filler, ultra-high-frequency ultrasound, hyaluronic acid, aesthetic medicine

## Abstract

Dermal hyaluronic acid (HA) fillers are used for nasolabial fold correction, but no study is still available on the use of ultra-high-frequency ultrasound (UHFUS) with 70 MHz probes for the evaluation of HA distribution and wrinkle amelioration. We selected 13 patients who received HA filler, evaluated before (Time (T) 0) and after injection (T1), and after 24 weeks (T2). The dermal thickness and distribution of HA were registered, as well as the Wrinkle Severity Rating Scale (WSRS), Global Aesthetic Improvement Scale (GAIS), and wrinkle 3D fullness. The UHFUS dermal thickness was increased by 11% for both sides at T1 and by 7.4% and 6.8% for the right and left side, respectively, at T2 (*p* < 0.01). The 3D wrinkle fullness showed a T1 increase (+0.59 cc and +0.79 cc for the right and left side, respectively) with a T2 maintenance of 45% of the T1 fullness (*p*-value < 0.001). The only clinical score significantly modified was WSRS, with a reduction of 56% at T1 and of 47.1% at T2 (*p*-value < 0.001). Our study then demonstrated the efficacy of UHFUS in the assessment of nasolabial fold correction, representing also the first multi-modal evaluation of HA persistence and its visual subsequent aesthetic results.

## 1. Introduction

The process of aging influences the appearance of the skin by producing both microscopic and macroscopic transformations that lead to a decrease in volume [1]. Various factors, including environmental conditions and personal habits such as smoking, play a role in skin aging and drive the subsequent alterations in the extracellular matrix of the dermis, resorption of bony structures, gravity, and a decrease in and redistribution of adipose tissue [2]. Additionally, a reduction in hyaluronic acid levels accompanies skin aging, which can expedite dehydration, loss of elasticity, and the formation of wrinkles. In recent decades, there has been a growing emphasis on appearance and the notion of beauty, leading to the expansion of the field of aesthetic medicine. This field is dedicated to modifying aesthetic appearance by addressing various conditions through both invasive and minimally invasive procedures, with some of the most recent ones also involving bioartificial injectable products [3]. With the emergence of advanced technologies, today’s physicians have the opportunity to address these changes with a variety of techniques.

Regarding non-permanent injectable products, hyaluronic acid (HA)-based dermal fillers are among the most widely used, due to their effectiveness and safety compared to permanent surgical cosmetic procedures [4]. HA is a non-sulfated glycosaminoglycan, widely distributed in connective, epithelial, and neural tissues; in the skin, it confers flexibility and hydration through its ability to bind and retain water molecules [5]. HA-based dermal fillers are used to restore volume loss, fill in fine lines and wrinkles, increase lip volume or define lip contour, and perform various procedures such as chin augmentation, nose reshaping, mid-facial volumization, lip enhancement, and wrinkle correction [6]. HA-based dermal fillers then represent an ideal aesthetic tool since they are safe and effective, biocompatible, easy to inject and distribute, and easy to remove if necessary [7]. This type of filler can be easily removed in the event of complications or dissatisfaction with the aesthetic result by injecting commercially available hyaluronidase into the concerned area [1]. The aesthetic result of HA dermal filler injections is closely related to the permanence of the product and its redistribution in the tissues. The literature provides evidence on the detected tissue integration in ultrasound (US) images as early as 1 month after injection [8]. Although there are several theories regarding the interaction between HA filler and tissue components, there is still little scientific evidence on the matter, and few data are available on the instrumental detection of long-term permanence of fillers [9]. High-frequency ultrasound (HFUS) is a non-invasive method for the analysis of skin and subcutaneous tissue composition that can detect tissue composition and potential changes [6]. HFUS provides clear imaging of the dermis and hypodermis layers, but it is limited in accurately visualizing the epidermis, except for the palmar and plantar surfaces. This restriction makes HFUS inadequate for detecting the smallest structures and capturing the finest details. In contrast, ultra-high-frequency ultrasound (UHFUS) operates at frequencies above 20 megahertz (MHz), significantly higher than the frequencies used in conventional US systems that typically range from 2 to 18 MHz; therefore, it is particularly valuable in the field of dermatology. One of the key advantages of UHFUS in dermatology is its ability to identify microscopic structures. It can effectively visualize sebaceous glands, hair erector muscles, and hair follicles, providing valuable insights into various dermatological conditions and facilitating accurate diagnoses [10]. By revealing these intricate structures, UHFUS enhances the understanding of skin pathologies and aids in determining appropriate treatment approaches. For instance, UHFUS has demonstrated its efficacy in the field of dermato-oncology, particularly in relation to melanoma, a type of skin cancer known for its poor prognosis and persistently underestimated incidence [11]. The literature data point out that US evaluation with a 70 MHz probe allowed the non-invasive measurement of melanoma thickness, which showed an excellent correlation with Breslow thickness [12]. Moreover, the use of UHFUS has shown a promising role in the evaluation of chronic ulcers [13] and in the pre-operative characterization of basal cell carcinoma [14].

To date, numerous clinical studies have also been published on the vascular, musculoskeletal, and intraoral applications of UHFUS [13]. However, there is a lack of studies regarding the use of UHFUS in the field of aesthetic medicine. The aim of our study was then to use, for the first time in the literature, a UHFUS probe of 70 MHz to evaluate the distribution pattern and permanence of an HA filler for nasolabial fold correction, correlating the results with a panel of validated aesthetic scores and clinical images obtained from the selected population.

## 2. Material and Methods

We performed a single-center prospective cohort study, involving a population of patients followed by the Unit of Dermatology, University of Pisa, Pisa (Italy), from May 2022 to December 2022. A total of 13 subjects were enrolled in this study, all with written informed consent. The eligibility criteria for enrollment in the study were as follows: participants had to be over 18 years of age and exhibit visible and moderate–severe nasolabial wrinkles. Throughout the duration of the study, patients were required to refrain from any cosmetic procedures involving the face, minimize exposure to UV radiation without using sunscreen, and use contraception if they were women of childbearing age. Individuals who had undergone cosmetic correction procedures within the past six months or permanent procedures were not included in the study. The exclusion criteria included significant skin diseases, trauma and genetic defects of the face, allergies or hypersensitivity to the product being tested, presence of neoplasms or severe clinical conditions such as neurological disorders, recent treatment with antithrombotic or antiplatelet drugs within the week preceding the study, pregnancy or breastfeeding, and a tendency to develop hypertrophic, atrophic, or keloid scars. All the selected patients presented with moderate to severe nasolabial folds and received an injective treatment with HA filler into each side of the nasolabial folds, ranging from 0.6 to 1 mL at a concentration of 25 mg/mL. In particular, the HA filler used in this study was a cross-linked sodium hyaluronate, of non-animal origin, stabilized in a phosphate buffer with pH 7. No anesthetic products were applied on the treated areas before filler injection. All the anamnestic information of our population was obtained from our electronic database. Clinical and UHFUS evaluations were performed at Week (W) 0, as well as before (T0) and after injection (T1). A third evaluation was then carried out at W24 (T24). Moreover, at W24, patients filled out a questionnaire on the side effects occurring in the 2 weeks following the injection. Aesthetic evaluation was based on the Wrinkle Severity Rating Scale (WSRS) and Global Aesthetic Improvement Scale (GAIS), calculated by a well-trained investigator who also obtained clinical images using a 3D image system, VECTRA^®^ H2 (Canfield Scientific, Inc., Parsippany, NJ, USA) and its vector analysis program, Markerless Tracking. Through 3D photographs, a differential volumetric assessment in cubic centimeters (cc) was obtained. US images were taken using a linear 70 MHz probe (B-MODE), which was positioned transversally at the midpoint of both nasolabial folds, perpendicular to the skin. Each image was analyzed by two experienced operators to assess quantitative dermal thickness, obtained by measuring in millimeters (mm) the distance between the dermo-epidermal junction and the subcutaneous tissue.

Categorical data were described with absolute and relative (%) frequency, while continuous data were summarized with mean and standard deviation. To compare repeated measures of the dermal thickness and WSRS variables, ANOVA for repeated measures was applied, while, to compare paired data of the GAIS and delta fullness variables, a *t*-test for paired data was performed. Significance was set at 0.05, and all analyses were carried out using SPSS v.28 software.

## 3. Results

### Population Features

Our population consisted of 12/13 females (92%) and 1/13 male (8%), with a mean age of 51.6 years (59.6–43.8). Overall, 3/13 patients (23%) had phototype 1 on the Fitzpatrick scale, 3/13 patients (23%) showed phototype 2, and 7/13 patients (54%) had phototype 3. A history of sunburn and clinical signs of photodamage were registered in 11/13 patients (84.6%). Furthermore, 8/13 patients (61.5%) were smokers or former smokers. The mean BMI was 22.9 (19.2–26.6). No comorbidities were reported by our population.

At baseline (Time (T) 0), the severity of pre-filler wrinkles measured by WSRS was 3.4 (±0.8) for both right and left wrinkles. The mean dermal thickness measured by UHFUS was 2.04 mm (±0.09 mm) and 2.05 mm (±0.1 mm) for the right and left nasolabial folds, respectively. The mean volume of filler injected in the right nasolabial fold was 0.51 mL (±0.20 mL), and that in the left was 0.52 mL (±0.18 mL). Following the injection (T1), the severity of wrinkles measured by the WSRS was 1.5 (±0.7) on the right and 1.5 (±0.5) on the left, thus recording a significant reduction of 56% (*p*-value < 0.001). The overall judgment of improvement in cosmetic appearance using the GAIS score, even if not statistically significant, registered an amelioration to 1.6 (±0.8) on the right and 1.7 (±0.8) on the left. On UHFUS, there was an increase in mean dermal thickness to 2.28 mm (±0.15 mm) on the right and 2.27 mm (±0.15 mm) on the left, registering an increase of 11% in both thicknesses that resulted statistically significant (*p*-value < 0.001). The average degree of diffusion of HA-based materials was 2.6 (±0.4) for the right side and 2.2 (±0.1) for the left side. Furthermore, the volume assessment performed through the 3D photographs showed a significant average increase of +0.59 cc (±0.33 cc) and +0.79 cc (±0.39 cc) for the right and left side, respectively (*p*-value < 0.01). At W24 (T2), the severity of wrinkles measured using the WSRS was 1.8 (±0.6) for the right wrinkles and 1.7 (±0.6) for the left wrinkles. Therefore, the severity of wrinkles at 6 months showed a reduction of 47.1% (*p*-value < 0.001). The overall judgment of the improvement of the aesthetic aspect using the GAIS score, even if not statistically significant, registered an amelioration to 1.7 (±0.8) on the right and 1.8 (±0.7) on the left, thus maintaining 94% of the result obtained at baseline, immediately after the injection of the filler. Moreover, UHFUS detected a mean dermal thickness of 2.19 mm (±0.14 mm) on both sides, with subsequent statistically significant thickness increases of 7.4% and 6.8% from the baseline for the right and left sides, respectively (*p*-value < 0.01). Furthermore, the volume assessment performed through the 3D photographs showed mean increases of +0.26 cc (±0.13 cc) and +0.36 cc (±0.21 cc) for the right and left sides, respectively, thus maintaining 45% of the result obtained at baseline, immediately after the injection of the filler (*p* < 0.001).

## 4. Discussion

Dermal fillers are widely used in aesthetic medicine with the main purpose of filling wrinkles, creating volume, and correcting age-related tissue loss [4]. A vast range of dermal fillers are available on the market, which differ in composition, duration of effect, ease of administration, complications, and limitations [15]. US is a non-invasive imaging method that allows a real-time evaluation of both pathological lesions and healthy skin, and it is also able to detect exogenous materials. Their echogenicity depends on their composition; tissue augmentation products with a predominantly hydrophilic component usually appear as anechoic areas, while synthetic materials, such as silicon oil or polymethylmethacrylate, tend to be hyperechoic [16]. HA is a polysaccharide composed of repeating units of disaccharides D-glucuronic acid and N-acetyl-D-glucosamine linked by a 28 glucuronidic β (1→3) bond, representing a fundamental component of the extracellular matrix, synovial fluid, and vitreous humor [17]. HA dermal fillers are widely used due to their physicochemical characteristics and biological properties, as well as their efficacy and easy management of adverse effects [18]. After the injection into the skin, HA initiates a mild inflammatory response at the interface with the host tissue. This initial reaction is subsequently succeeded by a progressive formation of fibrous tissue, which firmly secures the gel to the host tissue, effectively obstructing any displacement of the product [15]. However, the main limitation of this procedure is the limited permanence of the product (from 3 up to 12 months) [19], which affects the final aesthetic outcome. The longevity of HA fillers is determined by particle size, manufacturing processes, and consequent characteristics of the product, volume, location of injection, and host metabolism [15]. A previous study evaluated how dermal HA fillers could be observed and followed over time using a 25 MHz US probe and magnetic resonance imaging, pointing out the usefulness of the two combined techniques; however, the probe used had too low of a frequency to ensure optimal visualization of the dermis, thus necessitating another adjuvant imaging technique [20]. In another study, nasolabial wrinkles were evaluated with a 20 MHz probe after HA injection, which identified the injected HA as hypoechogenic or anechogenic areas that were well demarcated and homogeneous in the skin [6]. The study by Quiao et al. represented the first long-term evaluation of nasolabial HA filler and an important validation of the use of ultrasound for follow-up assessment [9]. Using a traditional 20 MHz probe, it is possible to identify the dermis and hypodermis, while the epidermis is poorly characterized. Traditional HFUS has limitations in accurately measuring skin thickness and identifying the dermal–epidermal junction [21]. On the contrary, the UHFUS 70 MHz probe precisely identifies the epidermis and obtains a more precise characterization of the dermis, attaining images with sub-millimeter precision of all the skin structures. UHFUS has demonstrated its ability to establish correlations between ultrasound images and histology with a high degree of specificity. This advanced technology enables the differentiation between healthy and pathological tissues, providing comparable accuracy to traditional biopsy methods. In particular, the epidermis of healthy patients displays a superficial hyperechogenic layer, an inferior hypoechogenic layer, and a hyperechogenic line delimiting it from the dermis [10]. The dermis appears as a hyperechogenic band, due to its collagen content, while the hypodermis presents hypoechoic fatty lobules and hyperechoic fibrous septae in between the lobules [22]. Thanks to the sub-millimeter precision and high resolution of the UHFUS images, we were able to determine the presence and distribution of the HA dermal filler and to determine the dermal thickness after injection and 6 months later (Figure 1 and Figure 2). Immediately after the injection of HA filler, a thorough examination of the ultrasound images revealed the presence of an inhomogeneous region within the treated area. This region displayed multiple anechoic oval areas, indicating the presence of fluid-filled cavities or voids. The overall appearance was characterized by an uneven distribution of the filler material. However, as the post-treatment period progressed, significant changes were observed over the course of 24 weeks. The previously identified inhomogeneous region began to exhibit a more uniform and homogeneous pattern. The anechoic oval areas, which had initially appeared scattered and unevenly distributed, gradually merged and became less pronounced (Figure 1 and Figure 2). Multiple descriptions of ultrasound images following the injection of HA can be found in the existing literature. However, it is important to note that these studies were conducted using lower-frequency probes, which can account for the observed discrepancies between our study and the previously reported data. Urdiales-Gálvez F et al. reported a globular distribution of HA filler at the ultrasound evaluation (12–18 MHz) performed after the injection and a heterogeneous pattern composed of alternating anechoic/hyperechoic after one month [23]. The ultrasound analysis conducted by Jiang involved the examination of 94 patients who received nasolabial fold filling [24], and ultrasound imaging was performed using 15 MHz frequency. In the study, HA dermal filler was predominantly observed as an anechoic structure with a distinct boundary, efficient sound transmission, consistent internal echo, and the absence of noticeable blood flow signals. The authors noted that, when HA dispersed into the surrounding tissue, it typically manifested as a hypoechoic structure. These anechoic or hypoechoic regions displayed an irregular distribution within the layered tissue, often forming a grid-like or honeycomb-like pattern. The different uses of US technology within the realm of aesthetic medicine have been explored in the literature, with one notable area of focus being the examination of US characteristics associated with volume-enhancing products [16]. In particular, it has been confirmed that fillers containing a hydrophilic component typically exhibit hypo- or anechoic patterns, appearing as subcutaneous pseudocystic deposits. This literature review also showed that the manifestations of face-filling materials under US are inconsistent due to different anatomical injection sites, different product characteristics, and, above all, different ultrasound probe frequencies. We can conclude that our data are comparable to those already found in the literature, and that the differences we observed depend mainly on the possibility of obtaining more detailed images thanks to the higher resolution of the 70 MHz probe.

Our study then represents further proof of the role of UHFUS as a non-invasive method in the field of aesthetic medicine, due to its ability to obtain precise dimensional evaluations and high-resolution images [25]. The potential applications of UHFUS in the field of cosmetic medicine remain largely unexplored. For example, one potential area of implementation involves the prospect of conducting echo-guided procedures using UHFUS. There are cases in the literature of complications from HA fillers treated with UHFUS-guided hyaluronidase injection, which ensured the safety of the procedure and the possibility of real-time monitoring [26]. Larger studies are needed to prove the superiority of the UHFUS-guided procedure. Moreover, UHFUS is considered as a valuable diagnostic tool in many fields other than dermatology, such as oral medicine and musculoskeletal anatomy [27].

To date, there is a lack of published research specifically focusing on the use of UHFUS for assessing the long-term durability of HA fillers in the skin. However, our study aimed to address this gap and provide insights into the persistence of HA fillers over an extended period. In our investigation, we were able to identify the presence of HA filler six months after injection using UHFUS. This finding is consistent with another study which used a 20 MHz probe to examine the longevity of HA filler six months post injection [9]. Their results align with ours, indicating that HA fillers can maintain their presence in the skin for at least half a year.

Furthermore, Fino et al. conducted a comparative analysis of two types of HA fillers for the treatment of nasolabial wrinkles. Their research determined that the average duration of both products was approximately 9.5 months [28]. This evidence suggests that HA fillers can provide noticeable effects for a considerable period. In contrast, Kalmanson et al. reported a particular case of a patient who exhibited persistent HA filler in the zygomatic area 2.5 years after the initial injection. The persistence was confirmed through magnetic resonance imaging (MRI) [18]. This exceptional case underscores the possibility of HA fillers demonstrating a prolonged presence in specific circumstances. Nevertheless, its important to note that these findings are based on a limited number of studies with relatively small sample sizes and varying follow-up durations. Consequently, further research involving larger and more diverse populations, as well as longer follow-up periods, is necessary to establish a precise understanding of the permanence and durability of these dermal fillers. UHFUS was not only used to assess the presence and distribution pattern of the HA filler, but also led to the quantification of dermal thickness at each follow-up visit. We registered a slight reduction in dermal thickness at 6 months compared to the measurement after the injection; however, dermal thickness remained higher than baseline values. We can consequently deduce that, 6 months later, the HA is integrated but still present. These results confirm the bio-integration of injected HA and its subsequent heterogeneous distribution, thus indicating degradation and reabsorption in the tissue [9,23]. Limited research exists in the literature that has specifically investigated dermal thickness through ultrasound following the administration of HA dermal filler. In a recently conducted study led by Bravo BSF, researchers examined the effects of a single session of a combined hybrid filler, consisting of HA and calcium hydroxyapatite, on dermal thickness. The study focused on 15 participants with mild to moderate sagging in the jawline, ranging in age from 32 to 63 years. US analysis was performed immediately before the procedure and at 30, 90, and 120 days post treatment. The assessment of dermal thickness in the preauricular regions was carried out using an HF-US device equipped with a linear 18 MHz probe. The ultrasound evaluation encompassed two areas: the treated area (right preauricular region) and a small untreated area on the left preauricular region, serving as the control in this study. After the results analysis, it emerged that the intra-patient comparison showed a significant increase in dermal thickness on the treated side. Furthermore, in the US evaluation conducted by Qiao et al. [9], dermal thickness was measured, and the results indicated an increase in dermal thickness and a lower echogenicity at 2 weeks and 24 weeks after the administration of HA filler. The decrease in dermal thickness at 48 weeks compared to 24 weeks post injection indicates the possibility of diffusion, reabsorption, degradation, or fragmentation of the fillers within the skin tissues. To obtain a clinical assessment of the severity of nasolabial wrinkles and the cosmetic improvement achieved, we used WSRS and GAIS. The WSRS is a validated clinical outcome instrument, widely used for the assessment of facial wrinkles and effectiveness of soft-tissue augmentation [29], while GAIS is a five-point scale ranging from much improved (1) to much worse (5). A previous meta-analysis by Stefura T et al. [30] regarding tissue filler for the nasolabial area reported a mean improvement of 1.21 in the WSRS score at 6 months and an increase from a GAIS score of 2.2 one month after filler injection to a score of 2.32 after six months. In our group, we registered a significant difference of 1.6 between baseline and 6 months follow-up in the WSRS score, reporting a substantial stability of the aesthetic outcome at 6 months, measured using GAIS. To visually evaluate both scores, we referred to 3D photographs of the patient’s face using a 3D image system, VECTRA^®^ H2 (Canfield Scientific, Inc. Parsippany, NJ, USA). By using the 3D images, we were able to achieve a level of precision and detail that traditional 2D imaging methods would not have provided, Moreover, the use of 3D images enabled us to conduct a volumetric assessment of the nasolabial folds and provided valuable insight into the maintenance of results over time. The data indicated that, after a 6-month period following the initial injection, 45% of the initially achieved results were notably still maintained (Table 1). This speaks to the effectiveness and durability of the injection, underlining its positive impact on the patient’s facial aesthetics. This additional method of evaluation proves very useful in aesthetic medicine, especially in procedures involving volume modification, which can, thus, be measured precisely and objectively. The relationship between UHFUS images and 3D imaging will require additional investigation in forthcoming research.

The overall view of the dermal thickness measured with UHFUS and the reduction in WSRS score can be appreciated in Figure 3.

This study had some limitations, including the inherent challenge of achieving complete standardization both in the injection process and in the imaging evaluation. Regarding the later aspect, to advance the application of ultrasound in aesthetic medicine, it is necessary to establish new, objective, and replicable evaluation scales. Over time, the study of larger patient cohorts will allow the development of ultrasound evaluation metrics to assess the efficacy of hyaluronic acid fillers. In fact, the sample size employed in this study was relatively small, which should be taken into consideration when interpreting the results. In future research, exploring the relationship between the dermal response to hyaluronic acid injection and individual factors, such as smoking habits and the skin’s history of sunburn, could provide valuable insights. This avenue of investigation holds potential for uncovering correlations and understanding how these factors might influence the dermis’s reaction to hyaluronic acid.

## 5. Conclusions

Our study represents the first experience in the assessment of nasolabial fold correction through an UHFUS with 70 MHz probe. The possibility of acquiring exceptionally high-resolution images presents the opportunity, for the first time in the literature, to identify anatomical landmarks and HA dermal filler with unparalleled precision, allowing for meticulous descriptions with an accuracy down to the millimeter scale. Moreover, it represents the first multi-modal aesthetic assessment of nasolabial folds amelioration using US evaluation of HA persistence, validated clinical scores, and 3D images of the subsequent visual aesthetic results. An additional novelty of this approach is the possibility of accurately visualizing the anatomical sites where fillers are injected, which opens up interesting prospects for future progress. These include the implementation of echo-guided filler injections, which can significantly minimize the risks of severe adverse events. Additionally, the ability to perform echo-guided dissolution with hyaluronidase becomes a viable option, further enhancing safety and accuracy in aesthetic procedures. It is crucial to emphasize the considerable potential for the deepening and expansion of the role of UHFUS in aesthetic medicine. To conclude, we propose the use of UHFUS for the follow-up of patients undergoing HA filler injections, due to the low invasiveness of the diagnostic technique and the high precision in detecting the deposition of the selected product.

## Figures and Tables

**Figure 1 diagnostics-13-02761-f001:**
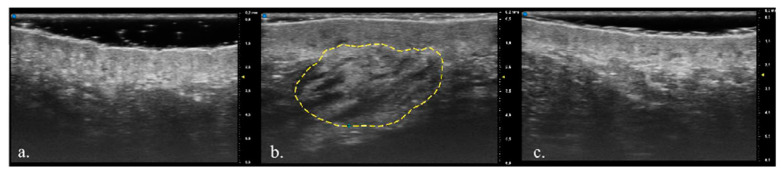
(**a**) Ultra-high-frequency ultrasound (UHFUS) of a 45-year-old patient’s nasolabial area before hyaluronic acid (HA) filler injection; (**b**) UHFUS of nasolabial area after HA filler injection (dashed yellow line), with the HA filler visible as anechogenic areas in the dermis; (**c**) UHFUS 6 months after the injection.

**Figure 2 diagnostics-13-02761-f002:**
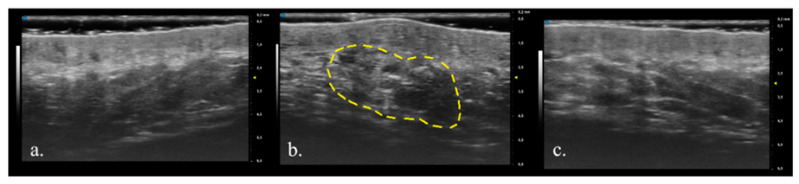
(**a**) Ultra-high-frequency ultrasound (UHFUS) of nasolabial area before hyaluronic acid (HA) filler injection, (**b**) after HA filler injection with the yellow dashed lines indicating the injected HA filler, and (**c**) 6 months after injection of a 58-year-old patient.

**Figure 3 diagnostics-13-02761-f003:**
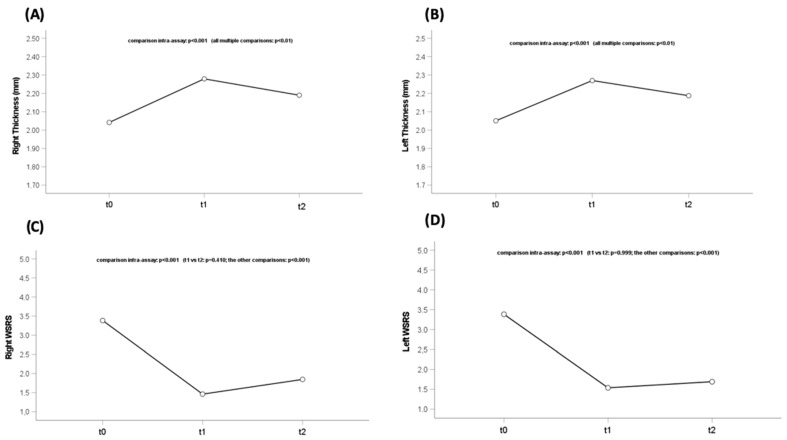
Dermal thickness modifications and statistical differences in the (**A**) right and (**B**) left naso-labial folds. WSRS modifications and statistical differences in the (**C**) right and (**D**) left nasolabial folds. WSRS: Wrinkle Severity Rating Scale.

**Table 1 diagnostics-13-02761-t001:** Delta fullness comparison. Statistics: mean, standard deviation (sd).

Delta Fullness	Mean	Sd	*p*-Value
Between T1 and T0 dx	0.59	0.34	<0.001
Between T24w and T1 dx	−0.47	0.41	
Between T1 and T0 sx	0.79	0.39	<0.001
Between T24w and T1 sx	−0.38	0.30	

## Data Availability

The data presented in this study are available on request from the corresponding author. The data are not publicly available due to privacy reasons.

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
