# Peer review of "Ultra-High-Frequency Ultrasound as an Innovative Imaging Evaluation of Hyaluronic Acid Filler in Nasolabial Folds"

_diagnostics, 2023, doi:10.3390/diagnostics13172761_

Round 1

Reviewer 1 Report

Thank you for the opportunity to read this interesting manuscript. I believe that the topic in aesthetic medicine is relevant and the wider use of ultrasound in HA imaging may have its uses. 

This paper has no exploratory aspect - it only presents the applications of skin ultrasound in the study of hyaluronic acid deposits in nasolabial folds. Of course, such studies can be performed, but the reason for using ultrasound must be concretized. It is obvious that HA deposits will be visible on ultrasound. In this aspect, it would be more important to study in 3D and to be able to correlate the 3D image - this needs to be highlighted more. 

Overall, the work is interesting, it presents new applications of ultrasound for evaluation - but here there should be created specific evaluation scales relating to HA imaging.  Perhaps in the future this will be covered more extensively.  

Minor concerns : 

Abbreviations - please spell out all abbreviations.

English needs requirements. 

In conclusion, the work is good, however, this essence of the novelty of the method is missing here - skin ultrasound and HA is now a trend that is lobbied for in the aesthetic medicine today. 

Spell check

Reviewer 2 Report

Very interesnting paper reflecting hard variables in relationship with the effect of HA  ( apart from volumization) in the sonographic features  of skin. Very well designed and clear. It would be nice to have  couple of pics  more in patient with differente aging features.
